



# Impact of local gravity wave forcing in the lower stratosphere on the polar vortex stability: Effect of longitudinal displacement

Nadja Samtleben[1], Aleš Kuchař[3], Petr Šácha[2,3,4], Petr Pišoft[3], and Christoph Jacobi[1]

[1]Institute for Meteorology, Universität Leipzig, Stephanstr. 3, 04103 Leipzig, Germany
[2]Institute for Meteorology, Universität für Bodenkultur Wien, Gregor-Mendel-Straße 33, 1180 Vienna, Austria
[3]Department of Atmospheric Physics, Faculty of Mathematics and Physics, Charles University, V Holesovickach 2, 180 00 Prague 8, Czech Republic
[4]EPhysLab, Faculty of Sciences, Universidade de Vigo, Campus As Lagoas, s/n, 32004 Ourense, Spain

**Correspondence:** Nadja Samtleben (Nadja.Samtleben@uni-leipzig.de)

**Abstract.** The effects of gravity wave (GW) breaking hotspots in the lower stratosphere, especially the role of their longitudinal distribution, are evaluated through a sensitivity study by using a simplified middle atmosphere circulation model. For the position of the local GW hotspot, we first selected a fixed latitude range between 37.5 and 62.5°N and a longitude range from 112.5 to 168.75°E , as well as an altitude range between 18 and 30 km. This confined GW hotspot was then shifted in longitude by 45° steps, so that we created 8 artificial GW hotspots in total. Strongly depending on the location of the respective GW hotspot with regard to the phase of the stationary planetary wave of wavenumber 1 (SPW 1) generated in the model, the local GW forcing may interfere constructively or destructively with the modeled SPW 1. GW hotspots, which are located in North America near the Rocky Mountains lead to an increase of the SPW 1 amplitude and EP flux, while hotspots located near the Caucasus, the Himalayas or the Scandinavian region lead to a decrease of these parameters. Thus, the polar vortex is less (Caucasus and Himalayan hotspots) or more weakened (Rocky Mountains hotspot) by the prevailing SPW activity. Because the local GW forcing generally suppresses wave propagation at midlatitudes, the SPWs 1 are propagating into the polar region, where the refractive index turned to positive values for the majority of the artificial GW hotspots. An additional source of SPW 1 may be local instabilities indicated by the reversal in the meridional potential vorticity gradient in the polar region in connection with a positive EP divergence. In most cases, the SPWs 1 are breaking in the polar region and maintain the deceleration and thus, the weakening of the polar vortex. While the SPWs 1 that form when the GW hotspots are located above North America propagate through the polar region into the middle atmosphere, the SPWs 1 in the remaining GW hotspot simulations were not able to propagate further upwards because of a negative refractive index above the positive refractive index anomaly in the polar region. GW hotspots, which are located near the Himalayas influence the mesosphere/lower thermosphere region because of possible local instabilities in the lower mesosphere generating additional SPWs 1, which propagate upwards into the mesosphere.



# 1 Introduction

Atmospheric dynamics is characterized by waves with different spatial and temporal scales (Douville, 2009) mainly forced in the lower part of the atmosphere, i.e. in the troposphere and stratosphere. One of the most important wave types, besides the planetary waves (PWs) and atmospheric tides, are gravity waves (GWs), which are maintaining the circulation and the thermal

structure of the upper atmosphere by exchanging energy and momentum and contributing to turbulence and mixing between all vertical layers (Fritts and Alexander, 2003). Their interaction with PWs (e.g. Manson et al., 2003; Jacobi et al., 2006; Hoffmann et al., 2012) or tides (Preusse et al., 2001; Beldon and Mitchell, 2010; Senf and Achatz, 2011, e.g.) can generate secondary waves, which may even influence the thermosphere (Miyahara and Forbes, 1991; Lilienthal et al., 2018). Thus, GWs are one of the main contributors into the coupling of different atmospheric layers. GWs are mainly generated by orography (Smith,

1985; Nastrom and Fritts, 1992), convection (Tsuda et al., 1994), jet sources (Plougonven and Zhang, 2014) or spontaneous adjustment processes (Fritts and Alexander, 2003), but not all of them are able to propagate into the middle atmosphere, which is strongly depending on their phase speed.

GWs exhibit a large spatial and temporal variability, which is closely linked to the synoptic conditions, the propagation conditions and the source of the GWs. Thus, there is a huge variability in the global GW distribution (Ern et al., 2004; Fröhlich et al.,

2007; Hoffmann et al., 2013; Schmidt et al., 2016). Most of the regions of enhanced GW activity are connected to (i) orography (Hoffmann et al., 2013), which are quite stable and persistent in space and time or (ii) to deep convection (in the Tropics, Ern and Preusse, 2012) as well as to jet sources (mainly in the midlatitudes, Plougonven and Zhang, 2014), which are spatially and temporally variable. The phase speed of nonorographically generated GWs (induced by convection, jet sources) is differing from the one of orographic GWs, which is highly influencing their propagation into the middle atmosphere (Andrews et al.,

1987). Depending on the position, strength and the induced wind shear of, e.g., the subtropical and polar front jet (jet sources), the generated GWs are either able to propagate into the mesosphere or break in the lower stratosphere (LS) (Gisinger et al., 2017). Apart from the phase speed also the amplitude of all kind of GWs is particularly important for the breaking conditions. Upward propagating GWs having large amplitudes, which increase the instability of the GWs, can already break in the LS (Fritts et al., 2016). These locally breaking GWs in the LS, which are limited in time, were already observed by Hoffmann

et al. (2013), Šácha et al. (2015) and Fritts et al. (2016). Anyhow, this leads to a transfer of momentum and energy, also called GW drag, on the local background flow in the LS and may also influence the stability of the polar vortex. In connection with an intensified PW activity, this process can produce a preconditioning of the polar vortex (Šácha et al., 2016; Samtleben et al., 2019) or even a sudden stratospheric warming (SSW) (Albers and Birner, 2014). Such an effect of breaking GWs in the LS has been also observed in several model studies (e.g. Plougonven et al., 2008; Constantino et al., 2015) as well as in satellite

measurements showing enhanced GW drag in the stratosphere before SSW events (Ern et al., 2016). Therefore, enhanced GW forcing in the LS may play an important role as a precursor and as an indicator of a potentially arising SSW.

Model experiments already showed that changes in GW parameters, e.g. the GW drag or the momentum flux, which modify the polar vortex geometry or even the stability (Samtleben et al., 2019), can lead to different kinds of vortex breakdowns (splitting or displacement) in connection with PW activity (Šácha et al., 2016; Scheffler et al., 2018). As a result, the vortex geometry





including a weakening or strengthening of the vortex itself strongly depends on the temporal and spatial GW drag distribution (Scheffler et al., 2018; Samtleben et al., 2019). These approaches provide a new basis regarding the evaluation of SSW events, which are strongly affected by GWs as well as by PWs. In an earlier study (Samtleben et al., 2019) we have analyzed the effect of an artificial GW hotspot on the polar vortex. The position was initially East Asia (EA) based on the results of Šácha et al.

(2015), and was shifted in latitude. To now provide information about more realistic distributions of GW hotspots, in this study we will displace the EA hotspot in longitude.

Thereby, we capture known GW hotspots like the Himalayas, the Rocky Mountains as well as Europe including several mountains. Furthermore, the displacement of the GW hotspot is along the polar front jet, so that also nonorographical GWs are considered. However, the chosen GW forcing of each artificial GW hotspot, which is the same for all GW hotspots in our sen-

sitivity study (see section 2.2 or Samtleben et al. (2019)), may not represent the corresponding realistic GW hotspot forcing. In reality, the GW forcing is strongly depending on the background conditions in the region of the respective GW hotspot, which is influenced by the prevailing stationary planetary wave (SPW) activity. Thus, the realistic GW forcing of each artificial GW hotspot would differ in strength as well as in direction, but this would complicate the interpretation of the results. For this purpose, our sensitivity study is more idealized and simplified. In the following section 2 of this paper, we will briefly

describe the used global circulation model (GCM), and will provide details on how the GW hotspots are implemented in this GCM. Section 3 describes and discusses the modeled dynamical effects of the GW hotspots on the circulation of the middle atmosphere, which includes the analysis of SPW activity and the wave propagation conditions. Section 4 concludes the paper.

## 2 Numerical model experiments

### 2.1 Model description and experiments

To analyze the middle atmosphere response to different local GW hotspots in the LS, we performed several experiments using the Middle and Upper Atmosphere Model (MUAM, Pogoreltsev et al., 2007; Lilienthal et al., 2017; Samtleben et al., 2019). MUAM is a mechanistic, 3D, nonlinear global circulation model, which extends in 56 layers up to an altitude of about 160 km in logarithmic pressure height $z$ with a vertical resolution $\Delta z = 2.842$ km. The logarithmic pressure height $z$ is defined by $z = -H \, ln(p/p_0)$ with a constant scale height $H = 7$ km and the reference pressure level $p_0 = 1000$ hPa. The deviation between

the logarithmic pressure height and the geometric height, which is strongly depending on the temperature profile, is small for altitudes up to 110 km with about 5 km. The zonal mean model temperature in the lowermost 10 km is nudged to 2000-2010 mean monthly mean ERA Interim (Dee et al., 2011) zonal mean temperature reanalysis data, which is necessary to correct the model climatology in the lower atmosphere, which is not included in the model. The lower boundary of the model at 1000 hPa is determined by 2000-2010 mean ERA Interim monthly and zonal mean temperature and geopotential reanalysis data as

well as by the corresponding extracted stationary planetary waves (SPWs) with wavenumbers 1-3. MUAM has a horizontal resolution of 5°/5.625° latitude/longitude. Radiative processes such as heating and cooling induced by absorption and emission are parameterized. The absorption of solar radiation by the most important atmospheric constituents such as $H_2O$, $CO_2$ and $O_3$ is realized according to Strobel (1986), based on prescribed water vapor and ozone fields.




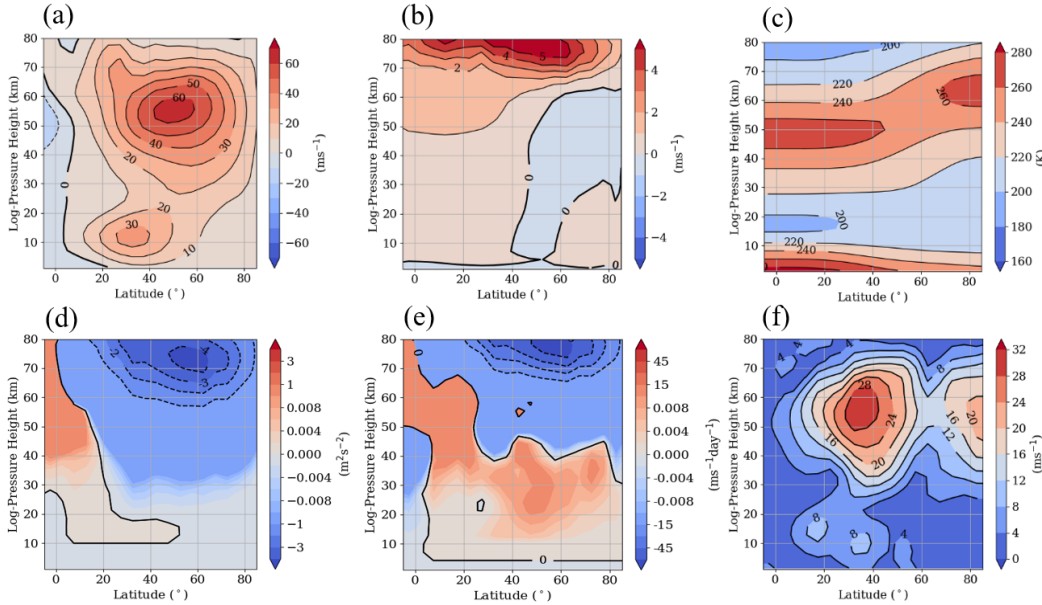

**Figure 1.** Zonal mean monthly mean (a) zonal wind (ms$^{-1}$), (b) meridional wind (ms$^{-1}$), (c) temperature (K), (d) zonal GW fluxes (m$^2$s$^{-2}$), (e) GW zonal wind acceleration (ms$^{-1}$day$^{-1}$) and (f) SPW 1 amplitude (ms$^{-1}$). Results refer to January conditions, and to the reference simulation.

Cooling due to infrared emission of O$_3$ in the 9.6 μm band and CO$_2$ are parameterized after Fomichev and Shved (1985) and Fomichev et al. (1998).

GWs are parameterized using an updated Lindzen-type linear scheme (Lindzen, 1981; Jakobs et al., 1986) with multiple breaking levels allowed (Fröhlich et al., 2003; Jacobi et al., 2006). The GWs are initialized at 10 km altitude, whereby at

each grid point 48 waves are initiated, which propagate in eight different directions. In each direction, GWs have six different phase speeds ranging from 5 to 30 ms$^{-1}$. The GW amplitudes are implemented as zonal means with a global average vertical velocity perturbation of 1 cm s$^{-1}$. The amplitudes are weighted using a prescribed latitude distribution, which is based on GW potential energy observations derived from GPS radio occultation measurements (Šácha et al., 2015; Lilienthal et al., 2017). More details on the model and the standard analysis procedures are given in Samtleben et al. (2019). Like in Samtleben et al.

(2019), we performed a reference simulation (Ref) for January conditions, using 2000-2010 mean ERA Interim reanalysis data for specifying the lower atmosphere dynamics. The Ref simulation results are shown in Fig. 1. Here, the January zonal mean latitude-height distributions of zonal (a) and meridional (b) winds, temperature (c), zonal GW flux (d), zonal wind acceleration due to breaking GWs (e), and the zonal wind SPW 1 amplitude (f) are presented. These results are the same as shown by Samtleben et al. (2019) and are repeated here for the sake of completeness and to facilitate the interpretation of the sensitivity

study results below.





## 2.2 Experiment description

GW breaking hotspots in the stratosphere lead to an additional energy and momentum transfer, which is connected to an increased GW drag. In order to simulate the effect of a GW hotspot, the zonal ($GWD_u$) and meridional ($GWD_v$) GW drag as well as the GW heating ($GWD_T$) have to be modified. We therefore enhanced the GW drag locally after the spin-up

period of the model (after 270 days) and let the model run for another 120 days as in the Ref simulation. On the basis of previous experiments performed by Šácha et al. (2016) we chose a moderate enhancement of -10 ms$^{-1}$day$^{-1}$ for $GWD_u$, -0.1 ms$^{-1}$day$^{-1}$ for $GWD_v$ and 0.05 Kday$^{-1}$ for $GWD_T$, which will not lead to a total breakdown of the simulated polar vortex. Owing to the nonlinear interactions between the background circulation and the GWs, the artificial GW forcing leads to changes in the background circulation, which in turn influences the GW propagation and breaking conditions, and consequently

modifies the GW drag and its regional distribution. This feedback mechanism was partly eliminated by turning off the GW parameterization in the further experiments and by using the GW drag output of the Ref simulation, which was modified in the GW hotspot region. As a starting point for our experiments as in Samtleben et al. (2019), we first focused on the observed Asian GW breaking hotspot, which was approximated by an enhanced GW drag between 37.5°N-62.5°N, 118.1°E-174.3°E and 18-30 km. This simulation is referred to as the H3 simulation. The $GWD_u$ distribution of the Ref (left) and the H3 (right)

simulation, averaged for the last 30 days of analysis (day 390-420), is shown in Fig. 2(a) for about 27 km altitude. During the analysis of the experiments, we are mainly focusing on steady states and therefore, mainly concentrate on the last 30 days of the simulations and neglect short term variabilities. In the Ref simulation, the $GWD_u$ varies between -0.02 and +0.02 ms$^{-1}$day$^{-1}$ in the H3 GW hotspot region. With the implementation of the H3 GW hotspot (Fig. 2(b)), the $GWD_u$ has risen up to -10 ms$^{-1}$day$^{-1}$, i.e. the additional GW forcing exceeds the maximum westward (negative) value of the Ref simulation by a

factor of 500. The mean $GWD_u$ within the H3 hotspot area of the H3 simulation is -10 ms$^{-1}$day$^{-1}$ and thereby, 3300 larger than the mean $GWD_u$ of the Ref simulation, which is about 0.003 ms$^{-1}$day$^{-1}$. With respect to $GWD_v$ and $GWD_T$, both are in maximum (mean) 5 (100) times stronger than those in the Ref simulation. Although these differences caused by the additional GW forcing seem to be quite large, the zonal GW forcing is still moderate compared to estimations from observations, which can exceed 40 ms$^{-1}$day$^{-1}$, and from GW parameterizations in this region (Šácha et al., 2018).

Based on the approach of Samtleben et al. (2019), we now extend the sensitivity study by displacing the observed Asian GW hotspot (H3) longitudinally around one latitude circle. We therefore fixed the latitude (37.5°N-62.5°N) and altitude (18-30 km) range as well as the longitudinal extent of 56.25° but varied the position of the GW hotspot from 22.5°E-78.75°E (H1) to 22.5°W-33.75°E (H8) in steps of 45°. The other artificial GW hotspots are labeled in between by H2 through H7. The position of the simulated GW hotspots can be seen in Fig. 2(a). Compared to the first sensitivity study of Samtleben et al. (2019) the

longitudinal displacement captures real GW hotspots more realistically, which may be orographically induced by the Rocky mountains (H5), the Himalaya (H2) or the European mountains (H8) or, which may be generated by jet sources in the polar front region (H1-H8). Because of the displacement along a fixed latitude belt, the size of the artificial GW hotspots remains the same in contrast to the latitudinal displacement in Samtleben et al. (2019). Thus, the forcings of the individual GW hotspots are similar and the effect of the GW hotspot only depends on the position.





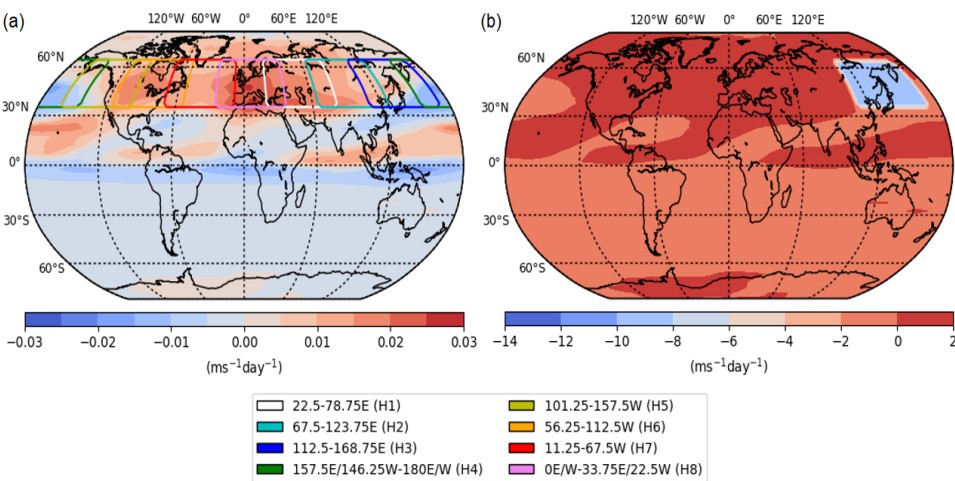

**Figure 2.** Zonal GW drag ($ms^{-1}day^{-1}$) at 26.9 km for the reference (a) and the H3 hotspot simulation (b). The last 30 model days are analyzed. Note the different scaling on the left panel.

## 3 Results

### 3.1 GW hotspot effect on the background circulation

To analyze the GW hotspot effects on the middle atmosphere dynamics, the zonal mean zonal wind and GW momentum flux differences between each GW hotspot H1-H8 (a-h) and the Ref simulation have been calculated. Both parameters, calculated
by considering the last 30 model days, are shown in a latitude-height plot only for the northern hemisphere in Fig. 3. Most of the local GW hotspots cause a deceleration of the west wind northward of 40°N up to the lower mesosphere. The effect is strongest for the H6 hotspot with 56.25-112.5°W longitudinal extent (Fig. 3(f)) with more than -20 m/s. The weakest effect is seen for the H1 hotspot at 22.5-78.75°E with only -4 m/s (Fig. 3(a)). However, the forcing is not strong enough to reverse the zonal mean zonal wind and to produce a major SSW.
Owing to the decreasing west wind, more eastward directed GWs, traveling faster than the background wind, can partly propagate into the middle atmosphere and counteract the deceleration of the dominating west wind. This is underlined by the GW momentum flux, which is less negative, i.e. showing a positive difference, in the regions of negative zonal mean zonal wind differences. The inversed effect is observed in regions of increasing west wind (positive zonal wind differences) for nearly all of the experiments, and particularly expressed for the H1 (22.5-78.75°E)-H3 (118.1-174.3°E), especially above 40 km altitude.
This shows an decreased impact of eastward directed GWs. Due to the increasing west wind in these regions more eastward directed GWs are filtered out (critical line) and the GW momentum flux becomes more negative. The negative/positive zonal wind anomalies at higher/lower latitudes mean that the polar vortex is strongly weakened and slightly shifted towards lower latitudes. The disturbance of the polar vortex can be also seen in the geopotential height and potential vorticity differences shown in polar plots in Fig. 4(a-h).



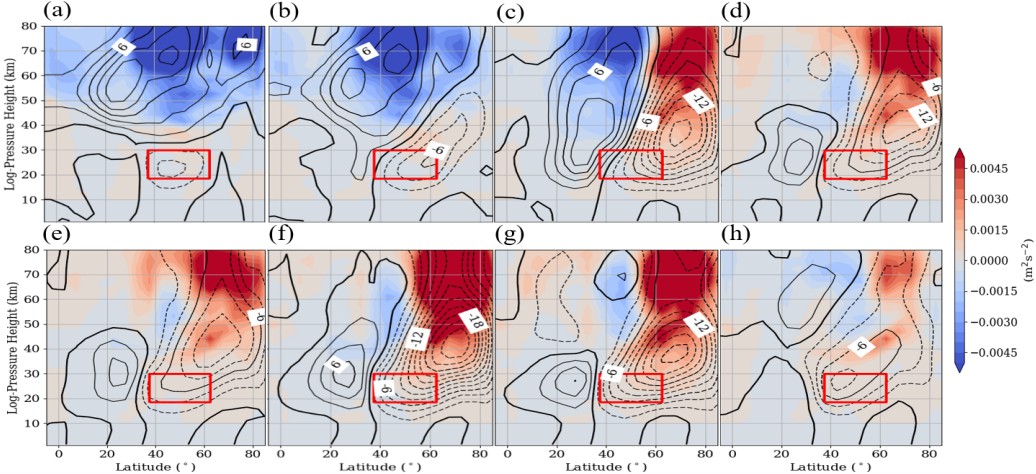

**Figure 3.** Zonal mean zonal wind (contour lines) and GW momentum flux (color coding) differences between the H1-H8 (a-h) and the Ref simulation. The last 30 model days are analyzed. Zonal wind differences are presented in intervals of 2 ms$^{-1}$, whereby the dashed (solid) lines represent negative (positive) differences. The zero line is highlighted. The position of the GW hotspots is shown by the red box.

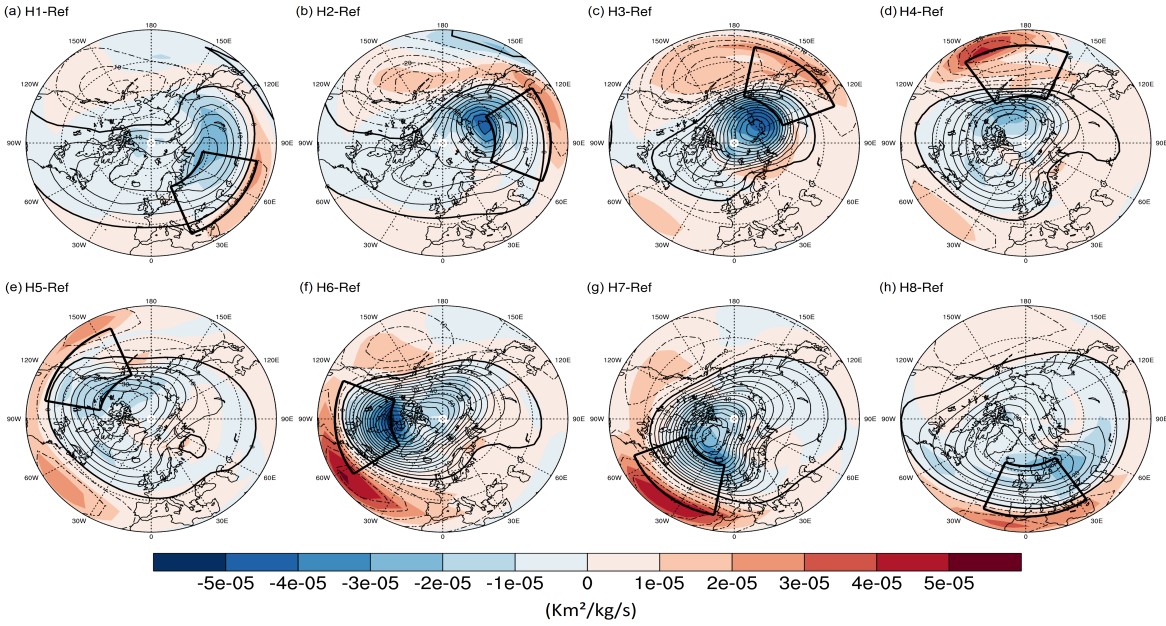

**Figure 4.** Polar plots of the geopotential height (contour lines) and potential vorticity (color coding) differences between the H1-H8 (a-h) and the Ref simulation averaged between 20 and 30 km. Latitudes ranges from 30°N to the pole. Geopotential differences have a highlighted zero line and are presented in intervals of 5 m, whereby the dashed (solid) line represents negative (positive) differences. The positions of the GW hotspots are illustrated by the black boxes.





Both parameters have been averaged over the 20 to 30 km altitude range. The potential vorticity differences are given in color coding and the geopotential height difference are illustrated by the contour lines in intervals of 5 m with a highlighted zero line. Again, the dashed (solid) lines represent negative (positive) differences. The position of each GW hotspot is illustrated by a black box. The potential vorticity combines the conservation of vorticity and mass in the atmospheric system as well as the

potential temperature for adiabatic processes. Because of the decreasing west wind and the destabilization of the polar vortex, the vorticity, which is normally increasing towards the polar region, is decreasing in each of the experiments. The decrease is strongest for the H3 (118.1-174.3°E) and H6 (56.25-112.5°W) GW hotspot in Fig. 4(c) and (f) and is mainly appearing at the northern flank of each GW hotspot. This is also the region of maximum geopotential height increase. Owing to the southward shift of the polar vortex, the potential vorticity is increasing in these regions. The increase is mostly occurring at

the southern flank of the GW hotspot. The distribution of the potential vorticity anomalies can be explained by means of the quasi-geostrophic potential vorticity $q_g$ equation, considering that meridional GW drag is negligible compared to the zonal drag, and neglecting diabatic processes:

$$\frac{D_g q_g}{Dt} \approx -\frac{\partial F_x}{\partial y} \tag{1}$$

with $F_x$ as the zonal GW drag. At the northern (southern) flank of the GW hotspot, $\frac{\partial F_x}{\partial y}$ is larger (smaller) than zero, so that $\frac{D_g q_g}{Dt}$ is smaller (larger) than zero, which may explain the negative (positive) potential vorticity anomalies at the northern (southern) flank of the respective GW hotspots. The displacement of the polar vortex is connected with an increase of the geopotential height. However, in the region of the Aleutian high pressure system, the geopotential height is decreasing. This effect can be observed for all GW hotspots and is most intense for the H3 (118.1-174.3°E) GW hotspot in Fig. 4(c). As a result

of the weakening of the Aleutian high, the polar vortex is less disturbed after the displacement towards lower latitudes. This corresponds to the strongest zonal mean flow increase at lower latitudes up to 40 km in Fig. 3.

### 3.2  Generation and propagation conditions of SPWs

The observed changes of the dynamics in the stratosphere and lower mesosphere and the related weakening of the polar vortex are mainly driven by the modulation of the SPWs. Fig. 5 shows the zonal wind SPW 1 amplitude and the EP flux and divergence

differences between each GW hotspot H1-H8 (a-h) and the Ref simulation results. The SPW 1 amplitude is strongly decreasing in the stratosphere and lower mesosphere for the H1 (22.5-78.75°E)-H3 (118.1-174.3°E) and the H7 (11.25-67.5°W) and H8 (33.75°E/22.5°W-0°E/W) GW hotspot, i.e. less SPWs 1 are propagating upward. This is in accordance to the decreasing EP flux and downward pointing arrows, underlining the fact, that the SPWs 1 propagate less in the middle atmosphere. The effect is strongest for the H1 (22.5-78.75°E) GW hotspot with a SPW 1 amplitude decrease of more than -12 ms$^{-1}$.

These regions of decreased SPW 1 amplitude are corresponding to regions of increased zonal mean zonal wind (Fig. 3). Due to the absent SPWs 1, less SPWs 1 are breaking with a consequently less transfer of momentum and energy, which decelerates the mesospheric jet. For this reason, we are observing a positive EP divergence difference showing that less SPWs are depositing their momentum. However, in comparison to the other GW hotspots, these GW hotspots (H1-H3, H7 and H8)

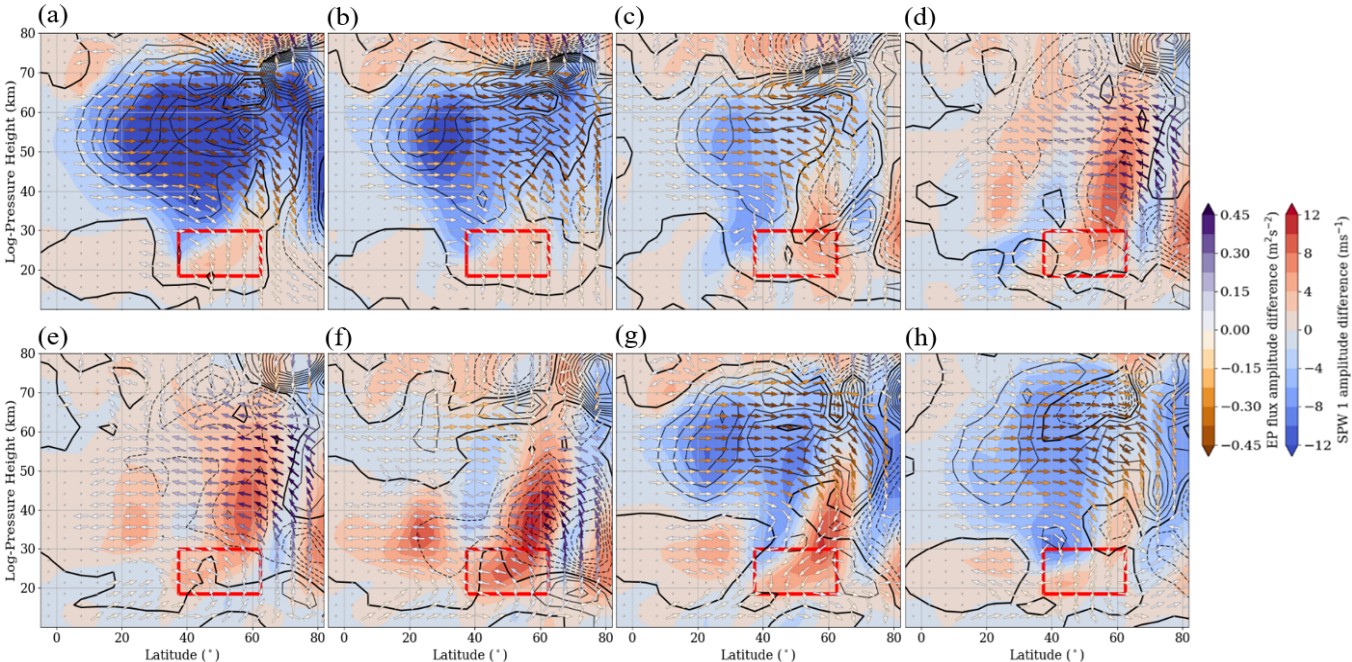

**Figure 5.** Zonal mean EP flux (arrows) and EP flux divergence (isolines, negative values are dashed) and zonal wind SPW 1 amplitudes (color coding) differences between all H1-8 (a-h) simulations and the reference simulation (H1-8 – Ref). The positions of the GW hotspots are shown by red boxes.

generate new SPWs 1 in the lower mesosphere (positive EP flux divergence), which propagate further upward and also affect the mesosphere/lower thermosphere (MLT) region. The source of SPWs 1 can be seen in the positive EP divergence difference, the increased EP flux and the arrows pointing upward. In case of the H4-H6 GW hotspots the EP flux has increased in the high-latitude stratosphere/lower mesosphere. The arrows are directed upwards northward of 50°N up to about 70 km altitude,

which means that more SPWs 1 are propagating into the middle atmosphere via the polar region. Some of these SPWs 1 are propagating into the middle atmosphere from the midlatitudes via the polar region, while some are directly generated in the polar region (positive EP divergence - source of SPWs 1). The increased SPWs 1 flux leads to increased SPW 1 amplitudes in the higher midlatitude stratosphere, which is strongest for the H6 (56.25-112.5°W) GW hotspot (Fig. 5(f)). Thus, more SPWs 1 are breaking, which amplifies the negative EP divergence and strongly decelerates the zonal mean flow. This is conform to the

results in Fig. 3 showing a decreasing zonal mean zonal wind at middle to high latitudes extending into the lower mesosphere. All simulations have in common that the SPW 1 amplitude is slightly decreasing at the southern as well as at the northern flank of each GW

hotspot. This effect is more apparent by plotting the SPW 1 amplitude for a specific altitude of about 35 km in Fig. 6. To investigate to what extent the SPWs 2 and SPWs 3 are modified by the additional GW forcing, we also added their amplitudes

in Fig. 6. As already shown in Fig. 5, the SPW 1 amplitude is strongest for the H6 (56.25-112.5°W, orange line) GW hotspot





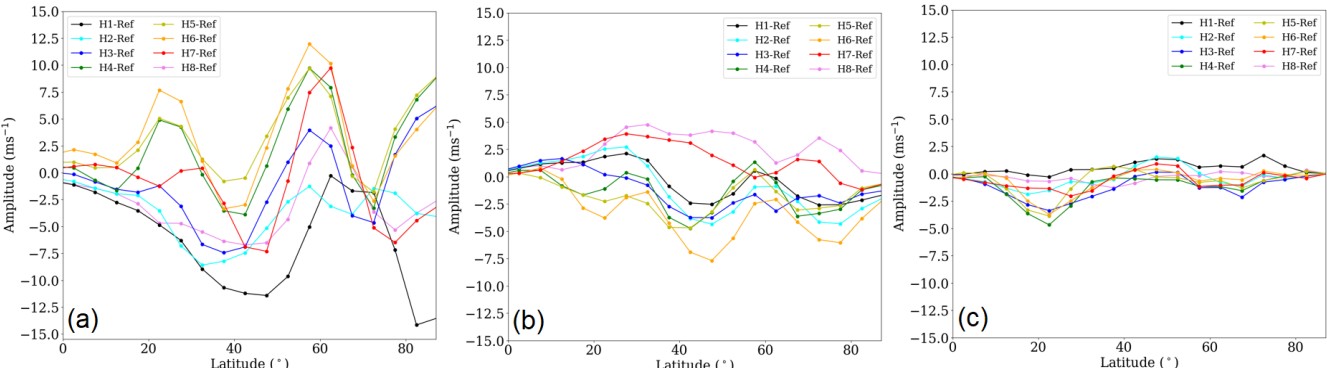

**Figure 6.** Zonal mean zonal wind SPW 1 (a), SPW 2 (b) and SPW 3 (c) amplitudes at 35 km altitude. Shown are differences between the H1-H8 and the Ref simulation.

with an increase of more than 12.5 ms$^{-1}$. The H1 (22.5-78.75°E, black line) GW hotspot exhibits the strongest decrease of the SPW 1 amplitude of more than -14 ms$^{-1}$. Regarding the SPW 1 anomalies owing to the GW hotspots can be seperated into three groups: (I) H1 (22.5-78.75°E) and H2 (112.5-168.75°E) (II) H3 (118.1-174.3°E) and H8 (33.75°E/22.5°W-0°E/W) and (III) H4-H7. The SPW 1 amplitudes of the first group (I) mainly decrease in the whole WH and only partly remain unchanged

around 60°N. The minima of the SPW 1 differences are located between 40 and 50°N and in the polar region. The second group (II) shows increased SPW 1 amplitudes at 60°N (less than 4 ms$^{-1}$) and decreased SPW 1 amplitudes (max. -7.5 ms$^{-1}$) around 40°N and 70°N. Although the SPW 1 anomalies of both hotspots are similar till 70°N, they diverge in the polar region, where the SPW 1 anomaly is positive for the H3 (118.1-174.3°E) and negative for the H8 (33.75°E/22.5°W-0°E/W) hotspot. Thus, the H8 (33.75°E/22.5°W-0°E/W) GW hotspot also exhibits features of the first group. The GW hotspots included in the

third group (III) lead to increased SPW 1 amplitudes between 20 and 30°N, around 60°N and in the polar region. Compared to the second group, the increase of the SPW 1 amplitude is much stronger in the respective region. Around 40 and 70°N the SPW 1 amplitude only slightly decreases except for the GW hotspot H7 (11.25-67.5°W) showing a stronger decrease of the SPW 1 amplitude (comparable to the second group). However, nearly all of the SPW 1 differences show the same pattern with decreasing amplitudes around 40 and 70°N. From Samtleben et al. (2019) we already know that this pattern is not caused by

the shape of the three dimensional box with a sharp transition zone of changed and unchanged GW drag values. This pattern can be also partly observed in the SPW 2 amplitude anomalies. Compared to the SPW 1 amplitude anomalies, the SPW 2 anomalies are less variable and cannot be really separated into different groups. Thus, the SPW 2 is less affected by local GW hotspots. Only in case of the H7 (11.25-67.5°W, red line) and H8 (33.75°E/22.5°W-0°E/W) GW hotspot (violet line) the SPW 2 amplitude has slightly increased with more than 4 ms$^{-1}$. Because the SPW 1 is highly dominating the middle atmosphere

dynamics in the H6 (56.25-112.5°W) simulation, the SPW 2 amplitudes is strongly reduced by about -7.5 ms$^{-1}$. For the H2-H5 GW hotspots, the SPW 2 anomalies show the same pattern as for the SPW 1 anomalies with decreasing amplitudes around 40 and 70°N (by less than -5 ms$^{-1}$). With respect to the SPW 3 amplitude anomalies, it can be observed that the SPW 3 is not massively influenced by the different GW hotspots except for the lower latitudes, where the SPW 3





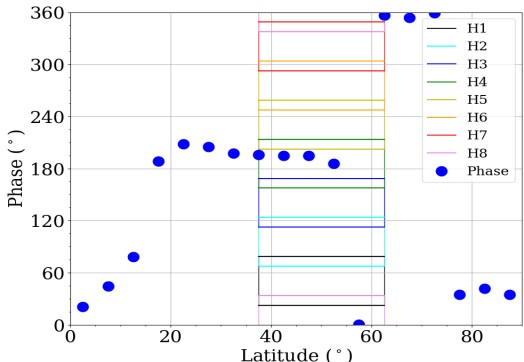

**Figure 7.** Zonal wind SPW 1 phase of the Ref simulation (blue dots). The positions of the GW hotspots are illustrated by the colored boxes.

amplitude is partly decreasing by more than 5 ms$^{-1}$ for the H3-H6 GW hotspots. To investigate if the local GW forcings, which can be interpreted as an additional wave 1, are in or out of phase with those in the model, in Fig. 7 the zonal wind SPW 1 phase from the Ref simulation at an altitude of 27 km is shown as blue dots. Since the GW drag is negative (westward), we again define here the SPW 1 phase as the longitude of maximum westward wind. The colored boxes in Fig. 7 represent the

longitudinal position of the local GW hotspots, here given in the range between 0 and 360°. The interference of two waves leads to a new wave with the same or larger (smaller) amplitude, if the phase difference of the two interfering waves ranges between 0 and +/- 120° (+/- 120 and 180°). For latitudes between 20 and 55°N, the phases of the SPW 1 ($\sim 200°$) and the local GW forcing are similar ($\sim 200° +/- 120°$), if the local GW forcing is active between 80° and 320° (in longitudes: 80°E-180°E and 40°W-180°W). This means in this latitude range, where the SPW 1 amplitude is usually largest (see Fig. 1(f)), the

GW hotspots are in phase with the existing SPW 1, which might therefore be enhanced. This would correspond to the GW hotspots H3-H6, which are completely located in this range. Between 0° and 80° as well as between 320 and 360°, the phase difference ranges between 120 and 240°, which means that SPWs 1, which will be forced by the GW forcing in this region interact destructively with the originally existing SPW 1. For this reason, GW hotspots such as the H1 (22.5-78.75°E) and H8 (33.75°E/22.5°W-0°E/W) ones, do not influence the middle atmosphere dynamics that much by additional SPWs 1 as those of

H3-H6. The H2 and H7 GW hotspots are partly in and out of phase with the in the model generated SPWs 1. This corresponds to (i) the results of Fig. 4 showing that the polar vortex is more strongly disturbed by the H3-H6 GW hotspots (ii) as well as to the EP flux and SPW 1 amplitude differences, which are negative for the H1 (22.5-78.75°E) and H8 (33.75°E/22.5°W-0°E/W) GW hotspots and less negative or even positive for the H3-H6 GW hotspots.

To investigate why SPWs 1 do not propagate into the middle atmosphere the refractive index $n$ (Matsuno, 1971; Andrews et al.,

1987) may be used. $n$ strongly depends on the meridional potential vorticity gradient $q_y$ and on the zonal mean zonal wind (Li et al., 2007). If $n > 0$ wave propagation is possible. The larger $n$ the higher is the probability that waves are propagating into the upper atmosphere. SPWs are attracted by the regions of a large, positive $n$. For this reason SPWs mainly propagate upward and towards the equator, where the zonal mean flow is weak. For $n < 0$ (east wind or strong west wind), the waves will break or will be reflected back to the troposphere and are not able to propagate (Matsuno, 1971). In Fig. 8 $n$ was multiplied





with the squared Earth radius. It shows the differences of $n$ between the H2 (112.5-168.75°E, (a))/H6 (56.25-112.5°W, (b)) and the Ref simulations. The zero line as well as the negative $n$ of the respective GW hotspot is indicated by the black line and the hatched area to show if negative differences refer to a negative or still positive $n$. In general, between 50 and 60°N, where the SPWs 1 are usually propagating upwards, $n$ is less positive or even negative (exceptions are H5 (101.25-157.5°W) and H8

(33.75°E/22.5°W-0°E/W)) in the region of the respective GW hotspot, as it can be seen for the cases H2 (112.5-168.75°E) and H6 (56.25-112.5°W) in Fig. 8(a) and (b). For this reason, less SPWs 1 are propagating via the midlatitudes into the middle atmosphere, which corresponds to the decreasing EP flux in this region. In the polar region, where we observe increased SPW 1 amplitudes (H1-H8) and EP fluxes (H4-H6), $n$ is increasing for all these simulations (not shown here). The effect is strongest for the H6 (56.25-112.5°W) simulation in Fig. 8(b). Thus, the suppressed SPWs 1 from the midlatitudes are able to propagate

into the polar region, i.e. the suppression is partly compensated. In most of the cases the SPWs 1 break there (negative EP divergence), because the propagation conditions are less suitable (H1-H3 and H7-H8). In turn, for the H4-H6 GW hotspots, the SPWs 1 are propagating from there further on into the middle atmosphere. While the SPWs of the H4 (157.5°E/146.24°W-180°E/W) and H5 (101.25-157.5°W) simulation are propagating up to 70 km, those of the H6 (56.25-112.5°W) simulation are just going up to 50 km. In contrast to the H4 (157.5°E/146.24°W-180°E/W) and H5 (101.25-157.5°W) simulation, the H6

(56.25-112.5°W) simulation shows a strongly negative $n$ anomaly (corresponding, in this case, to a negative $n$ - hatched area) above the positive anomaly in the polar region (Fig. 8(b)), which means that the SPWs cannot travel further upward. Compared to the other simulations, the H1-H3 and the H7 GW hotspots generate SPWs 1 in the lower mesosphere propagating into the MLT region. Generally, $n$ is negative in the upper mesosphere owing to the wind reversal (Samtleben et al., 2019). But due to the increasing west wind in this region, the propagation conditions change and $n$ becomes positive, which can be seen around

80 km between 60 and 75°N in case of the H2 simulation (Fig. 8a). Local positive EP divergence anomalies connected with an increase of the SPW 1 amplitude and originating EP flux underline the generation of SPWs 1 in the e.g. polar region or lower mesosphere.

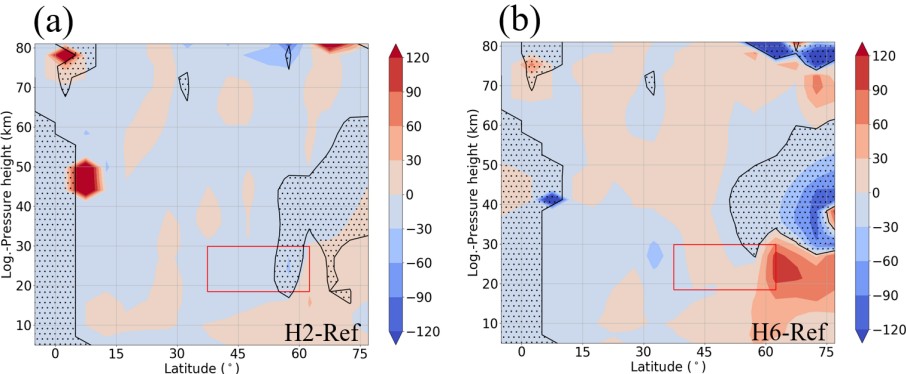

**Figure 8.** Zonal mean refractive index of SPW 1 differences between the H2 (a)/H6 (b) and the Ref simulations. The hashed regions denote regions of negative refractive index of the H2 and H6 simulation. The positions of the GW hotspots are illustrated by the red boxes.





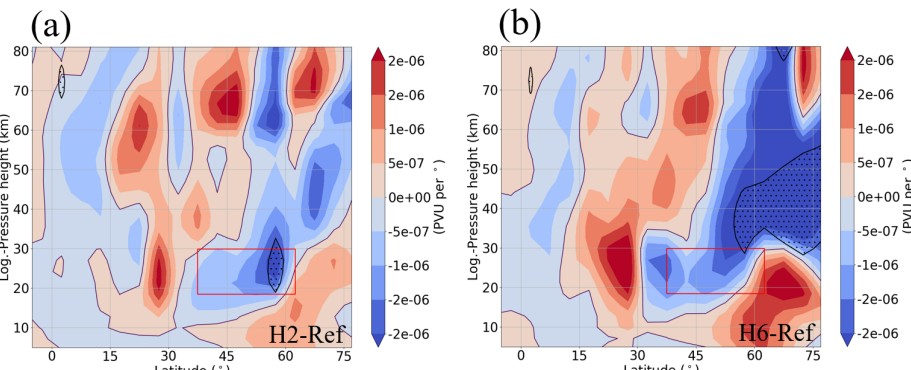

**Figure 9.** Zonal mean $q_y$ differences between the H2 (a)/H6 (b) and the Ref simulations, given in potential vorticity units (PVU) per degree. The hashed regions denote regions of negative $q_y$ of the H2 and H6 simulation. The positions of the GW hotspots are shown by the red boxes.

To see, whether this might be an effect of local instabilities, we analyzed $q_y$ differences between the H2 (112.5-168.75°E, (a))/H6 (56.25-112.5°W, (b)) and the Ref simulations (Fig. 9(b)). The red boxes represent the latitude-height position of the GW hotspots. In accordance to the results of Samtleben et al. (2019), $q_y$, which is usually increasing towards the polar region, reverses at the northern flank of each GW hotspot around 60°N due to the decreasing zonal mean zonal wind and the connected

weakening of the polar vortex. This negative $q_y$ anomaly (corresponding to a negative $q_y$ - hatched area) is strongest for the H6 (56.25-112.5°W) simulation in Fig. 9b corresponding to the strongest zonal mean zonal wind decrease. The local reversal of $q_y$ is an indicator for local instabilities (Charney and Stern, 1962) leading to the generation of nonstationary and stationary PWs. This may explain the locally enhanced SPW 1 amplitudes at the northern flank of each GW hotspot, which, however, are not able to propagate due to the unsuitable background conditions. While the negative $q_y$ as well as the anomaly of the H2 GW

hotspot are locally limited around 60°N, the one of the H6 simulation extends to the polar region. Thus, additional SPWs 1 are generated in the H6 simulation in the polar region (also enhanced positive EP divergence), where they are able to propagate further upwards and lead to the increased EP flux.

## 4  Conclusions

As an extension of the results of Samtleben et al. (2019) we performed a continuing sensitivity study, which investigates the

effect of breaking GW hotspots in the stratosphere and their impact on the middle atmosphere dynamics, which is mainly determined by the polar vortex. In contrast to Samtleben et al. (2019), we now concentrate on a fixed latitude range between 37.5 and 62.5°N and shift the artificial GW hotspot longitudinally in 45° steps starting with the observed breaking GW hotspot located between 112.5 and 168.75°E. Strongly depending on the position of the respective GW hotspot in relation to the phase of the SPW 1 generated in the model, the SPW 1 activity is either increasing or decreasing. Because the stability of the polar

vortex is affected by the prevailing SPW activity, we observe scenarios, in which the polar vortex is less or more weakened, i.e. the zonal mean zonal wind is decreasing in connection with increased SPW 1 EP fluxes and negative divergences. In





general, the local GW hotspots prevent the SPWs from propagating upwards at latitudes around 60°N, so that the SPWs propagate towards to polar region (positive $n$ (Karami et al., 2016)), where they are partly breaking and lead to the negative EP divergence decelerating the zonal mean flow. In some cases, also local instabilities indicated by the reversals in the $q_y$ generate new SPWs 1 in the polar region (Charney and Stern, 1962; Garcia, 1991), which cause an additional transfer of momentum

and energy. These local instabilities are also observed in the lower mesosphere, so that the SPWs 1 also have a lasting effect on the MLT region see (Smith, 2003; Lieberman et al., 2013; Matthias and Ern, 2018). For those GW hotspots, which are in phase with the modeled SPW 1 and do not create a negative $n$ in the upper polar region, the SPWs 1 propagate through the polar region further upward and influence the polar vortex up to an altitude of 70 km.

We briefly summarize the impact of each GW hotspot: The H1 (22.5-78.75°E) GW hotspot, which includes the Caucasus, the

Ural mountains and the Anatolian plateau as possible orographic GW sources produces an increase of zonal mean flow above 40 km and only a slight decrease below. The increase of the zonal mean flow is connected with a decrease of the SPW activity. This GW hotspot suppresses the SPW propagation at midlatitudes because it is out of phase with the SPW 1 originally generated in the model. It just slightly disturbs the polar vortex. The H2 (112.5-168.75°E) and H3 (118.1-174.3°E) GW hotspots, which include the Himalayas as possible orographic GW source (H2) and the Asian GW hotspot (H3), lead to decreasing (increasing)

zonal mean flow at middle to higher (low) latitudes, which is related to increasing (decreasing) SPW 1 amplitudes in the respective regions. Thus, the polar vortex is shifted southward. Both show a negative $n$ at midlatitudes preventing the SPWs from propagating upwards. The suppression of the SPWs is compensated by additional wave propagation towards the polar region (increase in $n$), where they are breaking and maintain the deceleration of the polar vortex. Besides the destabilization, here the polar vortex displacement, also the Aleutian high is weakened (decrease in geopotential height). The effect of the H7

(11.25-67.5°W) and H8 (33.75°E/22.5°W-0°E/W) GW hotspots, which include the Alps and the Scandinavian Mountains is similar. The H4-H6 GW hotspots, which include the Rocky Mountains show a strong increase in the SPW 1 activity causing a strong deceleration of the zonal mean flow at middle to higher latitudes. In contrast to the other simulations, the SPWs 1 can propagate into the middle atmosphere through the polar region, because they are in phase with the modeled SPW 1. The weakening of the Aleutian high pressure system is less intense for these three GW hotspots, so that the polar vortex is not

only disturbed by the enhanced SPW activity but also by the Aleutian high pressure system. Besides the specified orographic sources, all H1-H8 GW hotspots may also include convective or jet sources.

In all of these experiments, we have observed changes in the geometry or even a strong preconditioning of the polar vortex, which make the polar vortex in reality more vulnerable for enhanced SPW activity and may lead to a SSW. Several studies based on satellite observations or reanalysis data (Albers and Birner, 2014; Ern et al., 2016) already reported an increased GW

activity in terms of an enhanced GW momentum flux and GW drag before SSW events, which would fit to our basic idea for this sensitivity study. Because the effect is strongly depending on the longitudinal position of the GW hotspot in relation to the phase of the modeled SPW 1, in future studies we will also focus on the influence of different atmospheric phenomena such as ENSO or NAO. The distribution of the SPW 1 phase may be strongly different for each phenomenon, so that some of the artificial GW hotspots can be in phase with the SPW 1 and would cause different effects than we have observed now.

Additionally, instead of arbitrary positions of hotspots, their distribution should be more based on observations and known





stabel GW hotspots such as the Rocky Mountains or the Himalayas. Then, their analyzed impact and interaction will be more related to real atmospheric dynamical processes.

*Code availability.* MUAM model code is available from the corresponding author upon request.

*Author contributions.* N. Samtleben performed the model simulations and drafted the first version of the manuscript. P. Šácha provided the
5   GW potential energy climatology. A. Kuchař, P. Šácha, P. Pišoft, and C. Jacobi actively contributed to the discussions and the paper writing.

*Competing interests.* C. Jacobi is one of the Editors-in-Chief of Annales Geophysicae. The authors declare that there are no conflicts of interest.

*Acknowledgements.* N. Samtleben and C. Jacobi acknowledge support by Deutsche Forschungsgemeinschaft (DFG) through grant no. JA836/32-1. A. Kuchař, P. Pišoft and P. Šácha acknowledge support by GA CR through grants nos. 16-01562J and 18-01625S. ERA In-
10  terim reanalysis data have been provided by ECMWF through apps.ecmwf.int/datasets/data/.



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
