# Peer review of "Impact of local gravity wave forcing in the lower stratosphere on the polar vortex stability: Effect of longitudinal displacement"

_Annales Geophysicae, 2019_

## Referee Comment (RC1) · Anonymous Referee #1 · 25 Sep 2019

The paper "Impact of local gravity wave forcing in the lower stratosphere on the polar vortex stability: Effect of longitudinal displacement" by Samtleben et al. investigates the sensitivity of the northern hemisphere stratospheric polar winter jet to perturbations imposed by localized gravity wave forcing. This topic is of interest because localized forcing may be relevant for the jet evolution prior to sudden stratospheric warmings. In the fixed latitude range 37.5 to 62.5 deg N the locations of forcing are shifted in steps of 45deg longitude. It is found that forcing near the Rocky Mountains increases the stationary planetary wave 1 (vortex weakening), whereas forcing near the Caucasus, the Himalayas or the Scandinavian region decrease the wave 1 (vortex less weakened). Particularly, for forcing near the Himalayas it is found that local instabilities in the lower

mesosphere can generate additional stationary planetary waves that propagate into the mesosphere.

Overall, the paper is well written and publication of the paper in Annales Geophysicae is recommended with only minor comments.

Main comment:

One of the main results is that the position of the additional gravity wave forcing relative to the phase of the stationary wave 1 is of relevance, and the role of forcing at different locations is discussed. The effect of the different locations, however, strongly depends on the phase of the stationary wave 1 in the model. Therefore it should be discussed whether the phase of the simulated stationary wave 1 in the model is realistic.

Specific comments:

general comment: west wind -> westerly wind / east wind -> easterly wind

general comment: often the word "whereby" is used where it does not fit

p.2, l.16: persistant -> persistent

p.2, l.24: Here you state "which are limited in time," This statement is unclear. Did you want to say: "which occur only sporadically"?

p.4, caption of Fig.1: Please mention that only the Northern Hemisphere is displayed

p.5, l.7: please explain why GWDv is set negative

p.5, l.14: distribution of the Ref (left) and the H3 (right) -> distribution of the Ref (Fig.2a) and the H3 (Fig.2b)

p.5, l.15: is shown in Fig. 2(a) -> is shown in Fig. 2

p.6, l.10: decreasing west wind, -> weakening westerly wind,

p.6, l.13, l.15: increasing west wind -> strengthening westerly wind

p.8, l.30: of increased zonal mean zonal wind -> of strengthened zonal mean westerly wind ???

p.8, l.31/32: please check: which decelerates the mesospheric jet. -> and thus the zonal wind is less decelerated. ???

p.9, l.12 remove linebreak

p.10, l.4: please check: "WH" -> NH ??

p.11, l.5: 360deg -> 360deg East

p.11, l.5: interference -> superposition

p.11, l.23: (east wind or strong west wind) -> (easterly wind or strong westerly wind)

p.13, l.16: concentrate on -> focus on

p.15, l.1: stabel -> stable

---

## Author Comment (AC1) · 30 Sep 2019

Dear reviewer #1,

thank you for the comments and suggestions to improve the manuscript. Below we give some reply of the raised points, and we will carefully consider all of them in the revised manuscript.

**One of the main results is that the position of the additional gravity wave forcing relative to the phase of the stationary wave 1 is of relevance, and the role of forcing at different locations is discussed. The effect of the different locations, however, strongly depends on the phase of the stationary wave 1 in the model. Therefore, it should be discussed whether the phase of the simulated stationary wave 1 in the model is realistic.**

Thank you, this is a good and important point we did not consider in our discussion. The phases of the stationary planetary waves (SPWs) with wavenumber 1-3 are extracted from the 2000-2010 mean January mean ERA reanalysis temperature and geopotential data. Thus, the distribution of the SPW phases, which is included at the lower boundary of the model, is based on observations. To see if the model correctly reproduces the SPW 1 phases in the middle atmosphere, we compared them to SPW 1 phases extracted from SABER temperature measurements performed between 2002 and 2007 (mostly corresponds to our decadal mean) [Mukhtarov et al., 2010]. However, Mukhtarov et al. [2010] only provided a height-latitude cross section of the SPW 1 phase for December, while our simulations are based on January conditions. To see if we can still compare the datasets, we first had a look at the height-time cross section in Fig. 1 (a) showing the monthly averaged SPW 1 phase for the whole period (2002-2007) at 50°N [Mukhtarov et al., 2010]. We can see that the SPW 1 phase does not essentially change during December and January, so that they are more or less comparable (considering temporal averages of 6 years - might be different if the phases are presented separately for each year).

[Figure]

Figure 1: (a) Average (2002–2007) altitude-time cross section of the SPW1 phase at 50°N and average altitude-latitude cross sections of the SPW1 phase for (b) SABER (2002–2007) [Mukhtarov et al., 2010] and (c) MUAM (2000-2010).

By comparing the height-latitude distribution of the SPW 1 phases based on the SABER (b) and the MUAM (c) data in Fig. 1, it can be seen that the MUAM SPW 1 phases mostly correspond to the SABER SPW 1 phases. There are only small differences, which are possibly induced due to (i) the two different

time frames (2002-2007 for SABER and 2000-2010 for MUAM), and (ii) the different months (December for SABER and January for MUAM). Despite the small deviations, the SPW 1 phase seems to be quite realistic and creates a good basis for the analysis of the interference between the artificial gravity wave (GW) hotspots and the modeled SPW 1.

Reference:
Mukhtarov, P., D. Pancheva and B. Andonov, 2010: Climatology of the stationary planetary waves seen in the SABER/TIMED temperatures (2002-2007), J. Geophys. Res., 115, A06315, doi:10.1029/2009JA015156.

**west wind -> westerly wind / east wind -> easterly wind**

We will replace the words as suggested.

**often the word "whereby" is used where it does not fit**

Thank you for this remark. We will rephrase these paragraphs.

**p.2, l.16: persistant -> persistent**

We will correct this.

**p.2, l.24: Here you state "which are limited in time," This statement is unclear. Did you want to say: "which occur only sporadically"?**

We agree that it is a bit confusing. Yes, we meant sporadic or intermittent occurrence of breaking GWs in the lower stratosphere. We will change the sentence as suggested.

**p.4, caption of Fig.1: Please mention that only the Northern Hemisphere is displayed**

We will include your suggestion.

**p.5, l.7: please explain why GWDv is set negative**

Depending on the GW source and the background conditions, the GW drag can be separated into a zonal and meridional component. Compared to the meridional component, the zonal component is more pronounced. While it is already possible to derive the zonal GW drag from satellite measurements (still including large biases), the observation of the meridional GW drag is strongly constrained. So, the question was then how to estimate the local GW forcing of the observed breaking GW hotspot (in the East Asian/North Pacific region – H3 in the paper) and how to represent it within the model MUAM.
To get first an idea of the zonal and meridional GW drag direction, Šácha et al. [2015] analyzed the prevailing horizontal winds in the region of the observed breaking GW hotspot. Because the GW drag is acting against the zonal mean flow, the zonal and meridional GW drag were chosen according to the wind fields under the assumption that the GWs are orographically induced. As a result, both forcings were set to be negative.
The intensity of the zonal and meridional GW drag was examined in a previous sensitivity study, in which different kind of negative GW drag values were chosen [Šácha et al., 2016]. The setting with $GWD_u = -10ms^{-1}day^{-1}$, $GWD_v = -0.1ms^{-1}day^{-1}$ and $GWD_T = 0.05Kday^{-1}$ was a quite moderate GW forcing, which did not lead to total breakdown of the polar vortex. They also found that the strongest impact on the middle atmospheric circulation is caused by the zonal GW drag component, so that the meridional GW drag is more or less negligible (also the direction).

References:

Šácha, P., A. Kuchar, Ch. Jacobi, and P. Pišoft, 2015: Enhanced internal gravity wave activity and breaking over the Northeastern Pacific/Eastern Asian region, Atmos. Chem. Phys., 15, 13097-13112, doi:10.5194/acp-15-13097-2015.

Šácha, P., F. Lilienthal, Ch. Jacobi, and P. Pišoft, 2016: Influence of the spatial distribution of gravity wave activity on the middle atmospheric circulation and transport, Atmos. Chem. Phys., 16, 15755-15775, doi:10.5194/acp-16-15755-2016.

**p.5, l.14: distribution of the Ref (left) and the H3 (right) -> distribution of the Ref (Fig.2a) and the H3 (Fig.2b)**

We will correct this.

**p.5, l.15: is shown in Fig. 2(a) -> is shown in Fig. 2**

We will correct this.

**p.6, l.10: decreasing west wind, -> weakening westerly wind,**

We will change this sentence as suggested.

**p.6, l.13, l.15: increasing west wind -> strengthening westerly wind**

We will change this sentence as suggested.

**p.8, l.30: of increased zonal mean zonal wind -> of strengthened zonal mean westerly wind ???**

Yes, that's right. We will change this sentence as suggested.

**p.8, l.31/32: please check: which decelerates the mesospheric jet. -> and thus the zonal wind is less decelerated. ???**

Thank you for this comment. The sentence is also a bit confusing. We will rephrase it as follows:
Due to the absent SPWs 1, less SPWs 1 are breaking, which leads to a reduced transfer of momentum and energy, and thus, to a less decelerated zonal wind.

**p.9, l.12 remove line break**

We will remove it.

**p.10, l.4: please check: "WH" -> NH ??**

Thank you for this remark. We forgot to introduce this abbreviation. WH stands for winter hemisphere, which is on our case the Northern hemisphere. We will include that in the revised paper.

**p.11, l.5: 360deg -> 360deg East**

We will correct this.

**p.11, l.5: interference -> superposition**

We will replace the word as suggested.

**p.11, l.23: (east wind or strong west wind) -> (easterly wind or strong westerly wind)**

We will correct this as suggested.

**p.13, l.16: concentrate on -> focus on**

We will replace the word as suggested.

**p.15, l.1: stabel -> stable**

We will correct this.

---

## Referee Comment (RC2) · Anonymous Referee #2 · 16 Oct 2019

Review of "Impact of local gravity wave forcing in the lower stratosphere on the polar vortex stability: Effect of longitudinal displacement" by Nadja Samtleben et al.

General Comments

The manuscript by Samtleben et al. introduces a follow-up study on earlier work, assessing the impacts of idealized local gravity wave hotspots on the general circulation. In particular, the impact on the stationary planetary wave 1 in the northern hemisphere is analyzed. It is found that the different gravity wave hotspots can lead to both, weakening or strengthening of the polar vortex, depending on their location. Furthermore, the gravity wave hotspots may significantly affect the propagation of the stationary plan-

etary waves into the mesosphere compared to the reference run.

The study was carefully conducted and is scientifically sound, I think. The manuscript is very well written, clear, and concise. I would recommend it for publication in Annales Geophysicae, subject to a few minor comments listed below.

Specific Comments

p3, l20-33: Reading the description of the MUAM, I was wondering whether this is a somewhat simplified model (e.g., pretty coarse vertical resolution) and whether this possibly affects the results presented here?

p4, l9-15: Does the reference simulation agree well to reality? I assume it has been evaluated already in earlier studies, but it would be good if this would be stated here explicitly.

p14, l28-31: Although SSWs were not simulated in this study, I was wondering if you could say something about the timing between the gravity wave events and the following changes in the general circulation. How long did it take the circulation changes to manifest themselves?

---

## Author Comment (AC2) · 17 Oct 2019

Dear reviewer #2,

thank you for the comments and suggestions to improve the manuscript. Below we give some reply of the raised points, and we will carefully consider all of them in the revised manuscript.

**p3, l20-33: Reading the description of the MUAM, I was wondering whether this is a somewhat simplified model (e.g., pretty coarse vertical resolution) and whether this possibly affects the results presented here?**

Yes, it is true that the model is simplified. We intentionally used it since the purpose of the study was not to make detailed comparisons with observations, but to analyze the general response of the atmosphere to GW forcing. Using MUAM, we do not need to consider interaction of GW-PW processes with other dynamical features. It would not dramatically change our results if we would increase the resolution, because the scale of the considered processes is much larger. Model runs with refined resolution have shown an only small weakening of the polar vortex, so that our results would not be affected significantly.

**p4, l9-15: Does the reference simulation agree well to reality? I assume it has been evaluated already in earlier studies, but it would be good if this would be stated here explicitly.**

In the previous publication we compared parameters such as the zonal and meridional wind, temperature, stationary and planetary wave activity to climatologies and satellite observations. As a result, the model well reproduces the dynamics and processes in the middle atmosphere, and is therefore useful for the analyses of local breaking gravity wave hotspots. For the sake of completeness, we will shortly summarize the results from the previous paper and will include them in the revised manuscript.

**p14, l28-31: Although SSWs were not simulated in this study, I was wondering if you could say something about the timing between the gravity wave events and the following changes in the general circulation. How long did it take the circulation changes to manifest themselves?**

To identify the time interval, in which the atmosphere stabilizes after the gravity wave drag enhancement, in Fig. 1 the zonal mean zonal wind is shown in a latitude-time plot at an altitude of about 24km, so within the region of the forcing, where the effects should be strongest. As in the paper, we are now only concentrating on the H2 (a) and H6 (b) gravity wave hotspot. When we include the local gravity wave forcing after model day 270 and let the model run for another 120 days, we observe a shift of the polar vortex border from middle to lower latitudes. According to the results in the paper, the shift is more pronounced for the H6 GW hotspot.

[Figure]

Figure 1: Hovmöller diagram of the zonal mean zonal wind at 24 km for the H2 (a) and H6 (b) gravity wave hotspot.

The response to the additional GW forcing takes about 10-20 days, then the zonal mean zonal wind distribution nearly remains the same until model day 120. Because the temporal effect of such an artificial GW hotspot was already shown in the sensitivity study of Šácha et al. [2016, Fig. 3(c)], we will not include these figures in our revised paper, but we will shortly summarize the timing of the enhancement and the resulting circulation changes.

Reference:
Šácha, P., Lilienthal, F., Jacobi, C., and Pišoft, P.: Influence of the spatial distribution of gravity wave activity on the middle atmospheric dynamics, Atmos. Chem. Phys., 16, 15755–15775, https://doi.org/10.5194/acp-16-15755-2016, 2016.

---

## Author Response (AR1)

We are thankful for the reviewer's comments, which helped us to improve the paper. We have revised the paper according to the remarks, and hope that we sufficiently responded to each concern.

In the following we are using the abbreviations **P** and **L** for page and line.

We attach the revised version of the paper with changes marked.

**Comments of Reviewer 1:**

**One of the main results is that the position of the additional gravity wave forcing relative to the phase of the stationary wave 1 is of relevance, and the role of forcing at different locations is discussed. The effect of the different locations, however, strongly depends on the phase of the stationary wave 1 in the model. Therefore, it should be discussed whether the phase of the simulated stationary wave 1 in the model is realistic.**

The phases of the stationary planetary waves (SPWs) with wavenumber 1-3 are extracted from the 2000-2010 mean January mean ERA reanalysis temperature and geopotential data. Thus, the distribution of the SPW phases, which is included at the lower boundary of the model, is based on observations. To see if the model correctly reproduces the SPW 1 phases in the middle atmosphere, we compared them to SPW 1 phases extracted from SABER temperature measurements performed between 2002 and 2007 (mostly corresponding to our decadal mean) [Mukhtarov et al., 2010]. However, Mukhtarov et al. [2010] only provided a height-latitude cross section of the SPW 1 phase for December, while our simulations are based on January conditions.

[Figure]

Figure 1: (a) Average (2002–2007) altitude-time cross section of the SPW1 phase at 50°N and average altitude-latitude cross sections of the SPW1 phase for (b) SABER (2002–2007) [Mukhtarov et al., 2010] and (c) MUAM (2000-2010).

To see if we can still compare the datasets, we first had a look at the height-time cross section in Fig. 1(a) showing the monthly averaged SPW 1 phase for the whole period (2002-2007) at 50°N [Mukhtarov et al., 2010]. We can see that the SPW 1 phase does not essentially change during December and

January, so that they are more or less comparable (considering temporal averages of 6 years - might be different if the phases are presented separately for each year).

By comparing the height-latitude distribution of the SPW 1 phases based on the SABER (b) and the MUAM (c) data in Fig. 1, it can be seen that the MUAM SPW 1 phases mostly correspond to the SABER SPW 1 phases. There are only small differences, which are possibly due to (i) the two different time frames (2002-2007 for SABER and 2000-2010 for MUAM), and (ii) the different months (December for SABER and January for MUAM). Despite the small deviations, the SPW 1 phase seems to be quite realistic and creates a good basis for the analysis of the interference between the artificial gravity wave (GW) hotspots and the modeled SPW 1.

We corrected this on **P16L11**.

**Comments of Reviewer 2:**

**p3, l20-33: Reading the description of the MUAM, I was wondering whether this is a somewhat simplified model (e.g., pretty coarse vertical resolution) and whether this possibly affects the results presented here?**

The model is simplified. We intentionally used it since the purpose of the study was not to make detailed comparisons with observations, but to analyze the general response of the atmosphere to GW forcing. Using MUAM, we do not need to consider interaction of GW-PW processes with other dynamical features. It would not dramatically change our results if we would increase the resolution, because the scale of the considered processes is much larger. Model runs with refined resolution have shown an only small weakening of the polar vortex, so that our results would not be affected significantly.

We included a small paragraph on **P6L16-20 and P16L20-21** justifying that a refined model resolution would not significantly affect our results:
Potentially, it may also depend on the resolution but experiments with refined resolution (not shown here) did not significantly change the results because (i) we introduced the same GW forcing (same ratio of grid points with changed and unchanged GW drag), (ii) the circulation does not change dramatically (only a small weakening of the polar vortex) and (iii) we only consider large-scale processes in our analysis, which are not strongly affected by the resolution of the model.

**p4, l9-15: Does the reference simulation agree well to reality? I assume it has been evaluated already in earlier studies, but it would be good if this would be stated here explicitly.**

In the previous publication we compared parameters such as the zonal and meridional wind, temperature, stationary and planetary wave activity to climatologies and satellite observations. As a result, the model well reproduces the dynamics and processes in the middle atmosphere, and is therefore useful for the analyses of local breaking gravity wave hotspots. For the sake of completeness, we summarized the results from the previous paper and included them in the revised manuscript on **P5L1-4**:
As Samtleben et al. (2019) already mentioned, the model well reproduces the background parameters (zonal and meridional wind and temperature) in comparison to, e.g., CIRA-86 (Fleming et al., 1988) and URAP (Swinbank and Ortland, 2003) climatologies. Also, the GW flux and the SPW amplitude (slightly overestimated) distributions are similar to observations (Ern et al., 2016; Xiao et al., 2009).

**p14, l28-31: Although SSWs were not simulated in this study, I was wondering if you could say something about the timing between the gravity wave events and the following changes in the general circulation. How long did it take the circulation changes to manifest themselves?**

To identify the time interval, in which the atmosphere stabilizes after the gravity wave drag enhancement, in Fig. 1 the zonal mean zonal wind is shown in a latitude-time plot at an altitude of about 24km, so within the region of the forcing, where the effects should be strongest. As in the paper, we are now only concentrating on the H2 (a) and H6 (b) gravity wave hotspot. When we include the local gravity wave forcing after model day 270 and let the model run for another 120 days, we observe a shift of the polar vortex border from middle to lower latitudes. According to the results in the paper, the shift is more pronounced for the H6 GW hotspot. The response to the additional GW forcing takes about 10-20 days, then the zonal mean zonal wind distribution nearly remains the same until model day 120.

[Figure]

Figure 1: Hovmöller diagram of the zonal mean zonal wind at 24 km for the H2 (a) and H6 (b) gravity wave hotspot.

Because the temporal effect of such an artificial GW hotspot was already shown in the sensitivity study of Šácha et al. [2016, Fig. 3(c)], we did not include these figures in our revised paper, but we shortly summarized the timing of the enhancement and the resulting circulation changes on **P5L23-33**:

[revised manuscript text omitted]